# Coastal and regional marine heatwaves and cold-spells in the Northeast Atlantic

Amélie Simon[1*], Coline Poppeschi[2], Sandra Plecha[1],
Guillaume Charria[2], Ana Russo[1]

[1] Universidade de Lisboa, Faculdade de Ciências, Instituto Dom Luiz (IDL), 1749-016, Lisboa, Portugal
[2] Ifremer, Univ. Brest, CNRS, IRD, Laboratory for Ocean Physics and Satellite remote sensing (LOPS), IUEM, 29280 Brest, France

*corresponding author: Dr. Amélie Simon; ajsimon@fc.ul.pt

## Abstract

The latest IPCC report describes an increase in the number and intensity of marine heatwaves (MHWs) and a decrease in marine cold-spells (MCSs) in the global ocean. However, these reported changes are not uniform on a regional to local basis and it remains unknown if coastal areas follow the open ocean trends. Surface ocean temperature measurements collected by satellites (from 1982-2022) and 13 coastal buoys (from 1990-2022) are analyzed in the northeast Atlantic and three subregions: English Channel, Bay of Brest and Bay of Biscay. The activity metric, combining the number of events, intensity, duration and spatial extent, is used to evaluate the magnitude of these extreme events. The results from *in situ* and satellite datasets for each of the studied regions are quite in agreement, although the satellite dataset underestimates the amplitude of activity for both MHWs and MCS. This supports the applicability of the method to both *in situ* and satellite data, albeit with caution on the amplitude of these events. Also, this localized study in European coastal northeast Atlantic water highlights that similar changes are being seen in coastal and open oceans regarding extreme events of temperature, with MHWs being more frequent, longer, and extending over larger areas, while the opposite is seen for MCSs. These trends can be explained by changes in both the mean and variance of sea-surface temperature. In addition, the pace of evolution and dynamics of marine extreme events differs among the subregions. Among the three studied subregions, the English Channel is the region experiencing the strongest increase in summer MHWs activity over the last four decades. Summer MHWs were very active in the English Channel in 2022 due to long events, in the Bay of Biscay in 2018 due to intense events and in the Bay of Brest in 2017 due to a high occurrence of events. Winter MCSs were the largest in 1987 and 1986 due to long and intense events in the English Channel. Finally, our findings suggest that at an interannual time scale, the positive North Atlantic Oscillation favors the generation of strong summer MHWs in the northeast Atlantic, while dominant low-pressure conditions over northern Europe and a high off the Iberian Peninsula in winter dominates for MCSs. A preliminary analysis of air-sea heat fluxes suggests that, in this region, reduced cloud coverage is a key parameter for the generation of summer MHWs while strong winds and increased cloud coverage is important for the generation of winter MCSs.

**Keywords**

Extreme events, Sea Surface Temperature, Long-term *in situ* observations, Satellite data,
Marine heatwaves, Marine cold-spells, Bay of Biscay, English Channel, North Atlantic
Oscillation

## 1. Introduction
Heatwaves and cold-spells are extreme events in which there is a strong anomaly in
temperature for a certain period which can occur at a regional spatial scale. This type of
phenomenon can occur both in the atmosphere and in the ocean, with remarkable
consequences both for terrestrial and marine ecosystems (Ruthrof et al., 2018). In the case of
marine events (hereafter referred to as marine heatwaves (MHWs) or marine cold-spells
(MCSs)), severe large-scale biodiversity losses may occur such as species extinction, habitat
destruction and abrupt changes in the geographical distribution and structure of communities,
as well as the nutrient cycle (Frölicher and Laufkötter, 2018; Ruthrof et al., 2018; Smale et al.,
2019). Additionally, a decrease in the density of marine algae forests and coral bleaching
(Wernberg et al., 2016; Smale et al., 2019) have also been reported.

The frequency, duration and intensity of these extreme phenomena affecting ocean
systems have been increasing in recent decades (Lima and Wethey, 2012; Oliver et al., 2018;
Frölicher et al., 2018; Plecha and Soares, 2020; Simon et al., 2022 and many others). As a
result of global and regional warming heavily influenced by anthropogenic factors, the
intensity and annual number of MHWs will continue to accelerate globally (Oliver et al.,
2019; Plecha et al., 2021). Conversely, as oceans warm, MCSs are diminishing (Schlegel et
al., 2021; Simon et al., 2022) and are expected to continue to decline in the future (Yao et al.,
2022). However, these changes are not uniform regionally and it remains unknown if coastal
areas follow the open-ocean trends.

This paper focuses on the coastal and regional northeast Atlantic and three subregions
(English Channel, Bay of Brest and Bay of Biscay) as these zones are important for socio-
economic activities (e.g. fishery; Guo et al., 2022) and have contrasted dynamical
environments. Plecha et al. (2021) studied MHWs annual features in the whole north Atlantic
using low-resolution satellite data at 1º × 1º over the period 1971-2000. They show that in the
Bay of Biscay, the mean frequency is about 12 events per year and is characterized by ~ 12
days of mean duration and 0.4 ºC of mean intensity.  Marin et al. (2021) did a global analysis
of MHWs in coastal areas over the period 1992–2016 based on a multi-satellite product at a
resolution from 0.25°x0.25° to 0.05°x0.05°. They found that in the Bay of Biscay and English
Channel from 1992–2016, MHWs occurred on average 3 times per year lasting about 20 days
with a mean intensity of 1.5°C. Here we focus on these regions at the seasonal time-scale,
such as summer MHWs and winter MCSs using a satellite product at 0.25°x0.25°.
Long-term ocean warming is an important driver of the increase of MHWs (Frölicher
et al., 2018) and the diminishing of MCSs (Schlegel et al., 2021; Wang et al., 2022) but does
not explain shorter variabilities of these events, or their interannual variability. These marine
extreme events are also driven by other local and remote processes acting across a large range
of spatial and temporal scales (Holbrook et al., 2019; Schlegel et al., 2021). Modes of
atmospheric circulation variability can induce anomalous sea surface temperature (SST)
through modification of air-sea heat fluxes and/or displacement due to ocean current
advection (Deser et al., 2010) which for extreme cases, can lead to MHWs or MCS.
Interannual summer atmospheric variability in the north Atlantic-European sector has
been shown to be predominantly led by the summer north Atlantic Oscillation (SNAO)
pattern. The SNAO is identified as strong high-pressure anomalies dominating northern
Europe and weaker low-pressure over Greenland and the Iberian Peninsula which explains
about 20% of the variance using sea-level pressure (Hurrell et al., 2003). The SNAO is
recognized as a more northerly location and smaller spatial scale than the winter NAO pattern.
During the positive phase of the SNAO, northern Europe experiences drier, warmer and
reduced cloudiness conditions, and the Bay of Biscay, the English Channel, and the north and
Baltic Seas undergo warmer SST (Folland et al., 2009). The East Atlantic (EA) pattern is also
identified as a dominant mode of north Atlantic atmospheric variability (Barnston and
Livezey, 1987), which is particularly important for the northwest Iberian Peninsula climate in
all seasons (Lorenzo et al., 2008). It is a north–south dipole that spans the entire north Atlantic
Ocean, with centers southeastward to the NAO pattern (winter and summer).
Although there is strong evidence of the influence of large-scale features, no
consensus exists on atmospheric patterns associated with summer MHWs in the Bay of
Biscay and the English Channel. On one side, Holbrook et al. (2019) identify the Bay of
Biscay as a region for which there is a significant increase in annual MHWs days during a
positive phase of the NAO, based on a linearly detrended SST with satellite data and NAO
index. On the other side, Izquierdo et al. (2022a) suggest, based on the analysis of two *in situ*
buoys in the coastal south of the Bay of Biscay, that the incidence, duration, and intensity of
spring-summer MHWs is higher during the positive phase of the EA. However, for each of
these two studies, only one climate index out of the numerous modes of summer atmospheric
variability in the north Atlantic-Europe sector was considered.
MCSs have also been reported to occur as a response to atmospheric forcing through
anomalous winds and air-sea heat fluxes, especially in coastal regions where cold air
outbreaks over shallow water can cause rapid chilling of water (Crisp, 1964; Schlegel et al.,
2021). But to the best of our knowledge, no study has been published focusing on the
connection between MCSs and atmospheric circulation in the Bay of Biscay and the English
Channel.
At a more regional scale, Guinaldo et al. (2023) linked the 2022 MHW off France to
above-average solar radiation, below-average cloud coverage and negative wind speed
anomalies showing also the importance of hydrodynamic conditions such as the tide that
allows turbulent vertical mixing. This explains why the Mediterranean sea with weak tidal
ranges presents a more pronounced response to MHWs (Darmaraki et al., 2019; Simon et al.,
2022). Other studies have been carried out in terms of processes and detection of MHWs in
the Bay of Biscay but only along the Spanish Cantabrian coast. Namely, Izquierdo et al.
(2022b), found a steady increase in SST from 1998 to 2019, which was reflected in MHWs
occurrence and consequent match-up to report population shifts in coastal macroalgae. In a
second study, Izquierdo et al. (2022a) compared MHWs with satellite data and found a 6-fold
increase in their incidences in the last four decades with half of this increase related to climate
change.
Several studies focus on the impact of MHWs or MCSs on biological systems,
covering the areas of the Bay of Biscay, the English Channel or the Spanish Cantabrian coast,

reaching as far back as the 60s of the 20th century. These studies analyzed the atmospheric cold-spells of the winter of 1962-1963 on the English coast and the impact on marine animal mortality such as *Pecten Maximus* (Crisp, 1964) or migration of species such as flounder (Sims et al., 2004). In the English Channel, Gomez and Souissi (2008) made the link between the MCS of 2005 and the absence of the spring bloom of *Phaeocystis*. A delay in the initiation of the phytoplankton bloom caused by the presence of MCS at the end of winter in the Bay of Brest and in the Bay of Vilaine (in the northern part of the Bay of Biscay) is observed by Poppeschi et al. (2022). The impact of MHWs on biology is even more studied than the cold counterpart. Gomez and Souissi (2008) show the link between the heatwave of 2003 in the English Channel and the abundance of dinoflagellates. Joint and Smale (2017) demonstrate a link between MHWs and microbial activity assemblage in the English Channel which controls biogeochemical cycles in the ocean. The MHW of 2018 in the English Channel is present in the literature for its mortality mass impact on mussels (Seuront et al., 2019), its link to fucoids (Mieszkowska et al., 2020) or harmful phytoplankton blooms (Brown et al., 2022). Predictions at the atmospheric scale point to an increase in the future of heatwaves in the Bay of Biscay (Chust et al., 2011) and a decrease in marine fauna (Wethey and Woodin, 2022).

In this context, we aim to describe and explain the evolutions of the MHWs over summer and MCSs over winter activity in the northeast Atlantic and to investigate the regional variability in three subregions: the English Channel, the Bay of Brest and the Bay of Biscay. The analysis will rely on both *in situ* and satellite data to address MHWs and MCSs activity, aiming to evaluate the impact of such events in coastal regions and in the open ocean. This approach will allow us to evaluate the potential use of *in situ* measurements to detect, characterize and understand such extreme events. In the last section of this paper, we focus on the influence of the interannual atmospheric mode of variability involved in the occurrence of MHWs and MCSs in the northeast Atlantic by finding the atmospheric circulation occurring in phase with most of the strongest events.

## 2. Material and methods

2.1 Satellite and reanalysis data

The global SST data used in this study results from a combination of different observational platforms, including satellites, ships, buoys and Argo floats, provided by the National Oceanic and Atmospheric Administration (hereafter OISST; Reynolds et al., 2007; Huang et al., 2020). The satellite products have a daily temporal coverage for the 1982-2022 period and are interpolated to a regular global grid of 1/4º spatial resolution. Monthly geopotential height at 500 hPa (Z500), surface net short-wave radiation flux, surface net long-wave radiation flux, surface sensible heat flux and latent radiation heat flux data were obtained from the European Center for Medium-Range Weather Forecasts (ECMWF) reanalysis data ERA5 at a spatial resolution of 0.25°× 0.25° (Hersbach et al., 2019).

2.2 Buoy data

The *in situ* SST data are from autonomous coastal buoys that take continuous high-frequency measurements from 10 minutes to 1 hour (Figure 1, left panel). These buoys are from different European organizations, detailed below and in Table S1, covering the coastal areas of the English Channel, the Bay of Brest and the Bay of Biscay.

The National Observation Infrastructure network (COAST-HF, www.coast-hf.fr)
operates 14 buoys taking measurements of several physical and biogeochemical all around
French coasts. Among them, 7 buoys are used here and are located in the English Channel -
CARNot (https://doi.org/10.17882/39754), SMILe (https://doi.org/10.17882/53689) and
ASTAn; in the Bay of Brest - IROIse (https://doi.org/10.17882/74004) and SMARt
(https://doi.org/10.17882/86020) and in the Bay of Biscay - MOLIt
(https://doi.org/10.17882/46529) and ARCAchon. The Met Office (www.metoffice.gov.uk)
manages several buoys and also at offshore sites. The buoys used here are located in the
English Channel, on the South coast of England, SEVEn Stones; CHANnel and GREENwich.
The western Channel Observatory (WCO, www.westernchannelobservatory.org.uk), situated
within the western English Channel operates two oceanographic moorings. The station L4_Q
located near the city of Plymouth, approximately 7 km offshore is used here. Puertos del
Estado (www.puertos.es) operated two buoys along the Spanish coast: BILBao and GIJOn
located in the Cantabrian Sea, both of them are used here.

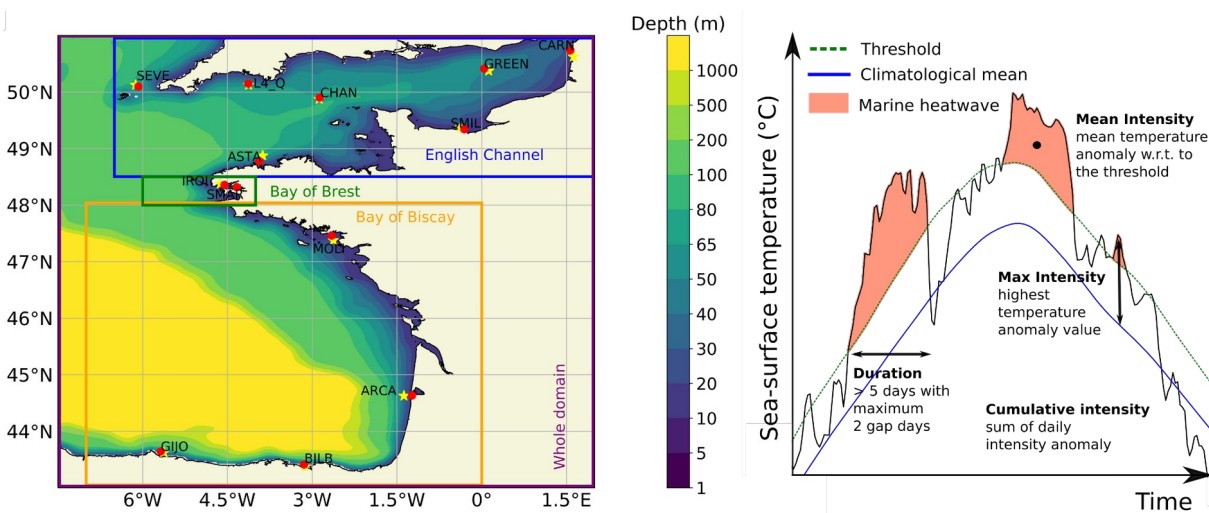

Figure 1: (Left) Map of the study area including the whole domain/northeast Atlantic (purple
box) as well as the three subregions which are the English Channel (blue box), the Bay of
Brest (green box) and the Bay of Biscay (orange box). The buoys are represented by red dots
and the closest satellite points are represented by yellow stars. (Right) Schematic of MHW
detection and properties as defined by Hobday et al. (2016).
2.3 Detection of MHWs and MCS
To detect marine temperature extreme anomalies, we use the definition of Hobday et
al. (2016). First, a climatology over 40 years, from 1982 to 2022, is calculated from the
satellite product. Then, we apply the $90^{th}$ percentile on summers (JJAS) for MHW and the $10^{th}$
on winters (DJFM) for MCS. Finally, a MHW (MCS) is detected if values are above (below)
the threshold for at least 5 days. For *in situ* data, the same detection method is applied
considering the climatology calculated from the satellite product. Only seasons (summer or
winter) with more than 80% of available data are analyzed.
To characterize MHW and MCS, we analyze parameters such as the number of events,
the duration, the spatial extent and the cumulative intensity, defined as in Hobday et al. (2016)
(Figure 1, right panel). We also explore an integrated indicator of these different parameters
characterizing the marine temperature extreme events (MHWs and MCS), called activity and
defined by Simon et al. (2022). This indicator estimates for each grid point the cumulative
combination of the mean intensity, the duration and the affected area of each extreme event
within a selective time range (for example JJAS). This activity index accounts explicitly for
the area, as in most SST products a grid cell area differs from one latitude to another and
marine thermal events can expand over large areas. The activity is calculated for each grid
point. It sums the product of the mean intensity, duration within the selected time range, and
area of each detected event occurring within the selected time range. The activity is written as
follows:
$$Activity = \sum_{EE \in Time\,Range} mean\,intensity_{EE} \cdot duration_{EE \cap Time\,Range} \cdot area_{EE}$$

Where EE ∈ time range, denotes the extreme event (EE) that occurs within the selected time
range; the mean intensity of EE (in ºC) is the mean temperature anomaly with respect to the
threshold of the event; duration EE ∩ time range (in days) is the duration of the event that
remains within the considered time range, and $area_{EE}$ (in km$^2$) is the area affected by the
discrete event within a predefined domain. Time series involving the activity metric for a
domain are calculated as the mean of every grid cell considered. The activity for each station
is computed in °C.days without considering the area influenced by the events as it can not be
estimated from single localized stations.
This method of detection and characterization of marine thermal extreme events is
performed over the whole domain of this study, referred to as the northeast Atlantic (8° W to
2° E - 43° N to 51° N) and at each station where *in situ* observations are available. As
illustrated in Figure 1, three different subregions will be analyzed in detail, namely (i) the
English Channel (6.5° W to 2° E - 48.5° N to 51° N), (ii) the Bay of Brest (6° W to 4° W -
48° N to 48.5° N) and (iii) the Bay of Biscay (7° W to 0° W - 43° N to 48° N). This will allow
us to explore these regions separately and highlight regional patterns. Those three subregions
can be associated with three contrasted hydrodynamical regimes: macrotidal (English
Channel), semi-enclosed bay (Bay of Brest), mesotidal (Bay of Biscay; Charria et al., 2013).

## 3. Results

3.1 Evolution of marine heatwave activity

3.1.1. An integrated regional view

MHWs were detected over the northeast Atlantic. The activity index (Figure 2a)
highlights two main periods in the MHWs dynamics. Before 2003, MHWs activity remained
moderate to weak with activity generally lower than 5 °C.days.10$^3$ km$^2$ corresponding to 1.2
mean occurrences per summer with a mean duration limited to 8 days (Figure 3). Only the
summer of 1989 displayed strong MHWs activity (exceeding 10 ºC.days.10³ km²) before
2000. From 2003 onward, the activity increased over 30 ºC.days.10³ km² for summers 2018
and 2022 associated with more than 2.5 mean occurrences lasting around 20 days. The mean
intensity remains quasi-steady during the whole period. The interannual variability and trend
of the summer MHWs activity for the whole domain is similar to the one obtained for the
average activity of the 13 grid cells closest to the buoy locations (black line of Figure 2),
suggesting that at first order of magnitude, coastal and open ocean follow the same evolution.

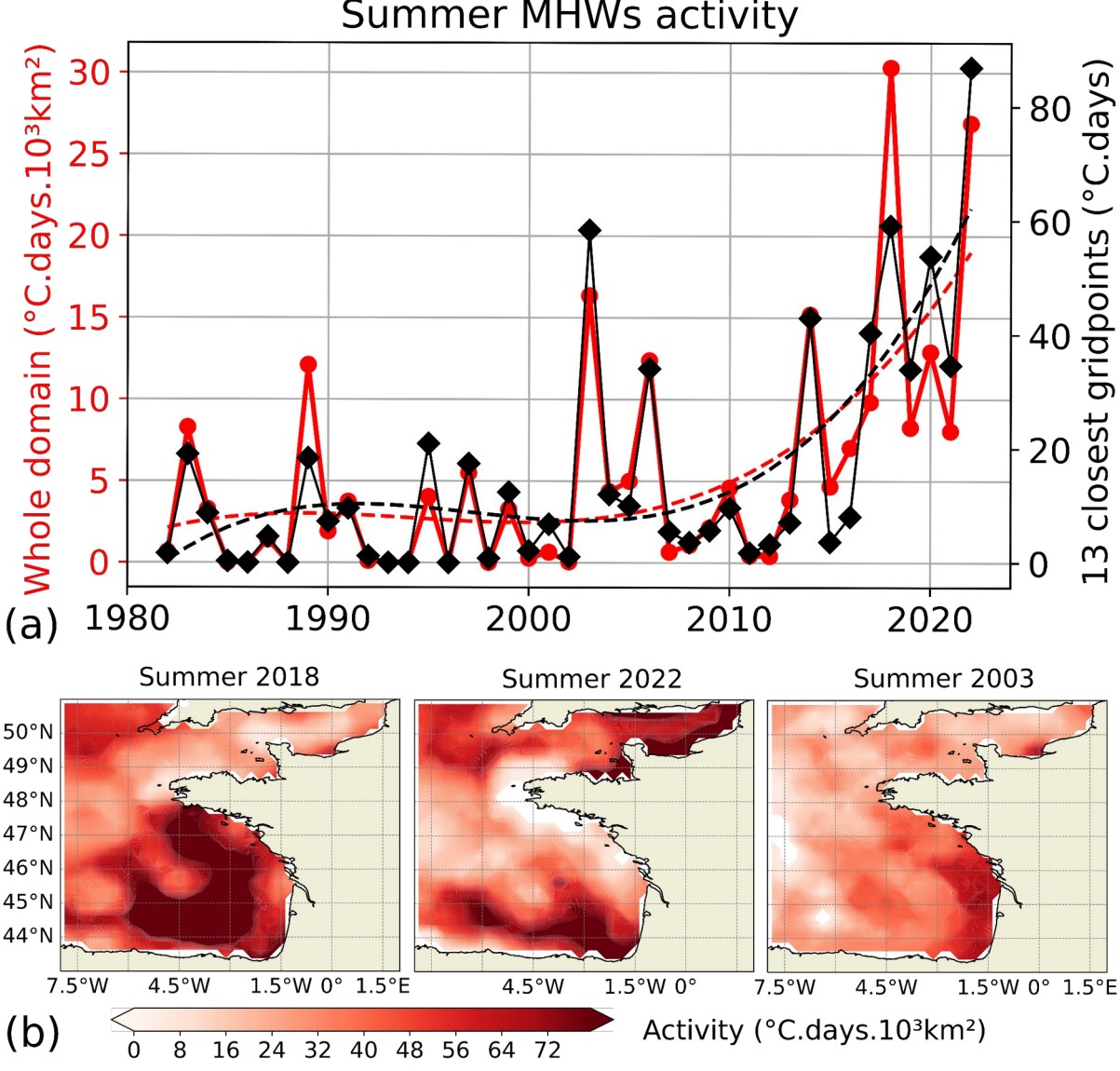

(a)

(b)

Figure 2: (a) Times series of summer (JJAS) MHWs mean activity in the northeast Atlantic from the satellite product (red curves with circle marks) and for the average of the 13 grid cells closest to the buoys from the satellite product (black curves with diamond marks). Dash lines represent the regression of a third-order polynomial of the solid line with the same color. (b) Summer (JJAS) activity (first row; in ºC.days.10³ km²) for the top 3 summers in terms of activity in the northeast Atlantic (from left to right).

The three most active summers are 2018, 2022 and 2003 (Figure 2a). During 2018 (Figure 2b), maximum activity is located in the Bay of Biscay over the outer continental shelf and the continental slope from the southern part of the Biscay. These events are also extending to the north to southern of Brittany and is limited by the Ushant tidal front (Le Boyer et al., 2009; Müller et al., 2010). Regions of minimum activity during 2018 are west of French Brittany in the Ushant front region where tides are efficiently mixing the water column. Similarly, the activity remains weak in the English Channel, as it is a macrotidal region. In terms of duration, longer events are observed in the southern part of the Bay of Biscay exceeding 30 days (Figure S1). The 2022 summer is the second most active year for the whole domain, with over 25 °C.days.10³ km², and also the strongest in terms of marine activity over coastal regions as shown by the maximum value of the average activity near the 13 buoys considered (Figure 2a). Spatially, the English Channel and the north of Spain record

the strongest MHWs activity while the French Brittany coast has no occurrence over this year
(Figure 2b). In the English Channel, the mean duration of the summer 2022 events was
around 35 days (Figure S2) with localized events lasting more than 50 days (Figure S1). In
northern Spain, the duration of the events was around 20 days, however, they occurred very
frequently over the summer with strong mean MHWs intensities of around 2 ºC (Figure S1).
In 2003 (Figure 2c), the MHWs activity spatial distribution was different than in 2018 and
2022. The activity is larger over the inner continental shelf along western French coasts in the
Bay of Biscay. This region is under the influence of significant river plumes along this coast
(Adour, Gironde and Loire rivers). During this year, river discharge could have induced
stratification (inducing faster warming of the surface mixed layer in regions of freshwater
influence; Oh et al., 2023) and warmer waters from rivers suggest that observed MHWs were
sustained by an atmospheric event more centered over lands. During this summer, the number
of events is larger in the western English Channel but shorter and less intense than in the Bay
of Biscay. These top three active summers highlight the interannual spatial variability of
MHWs activity. The detailed mean features (number of events, duration and mean intensity)
of summer MHWs over the period 1982-2022 in the northeast Atlantic, English Channel, Bay
of Brest and Bay of Biscay are documented in Table S2.

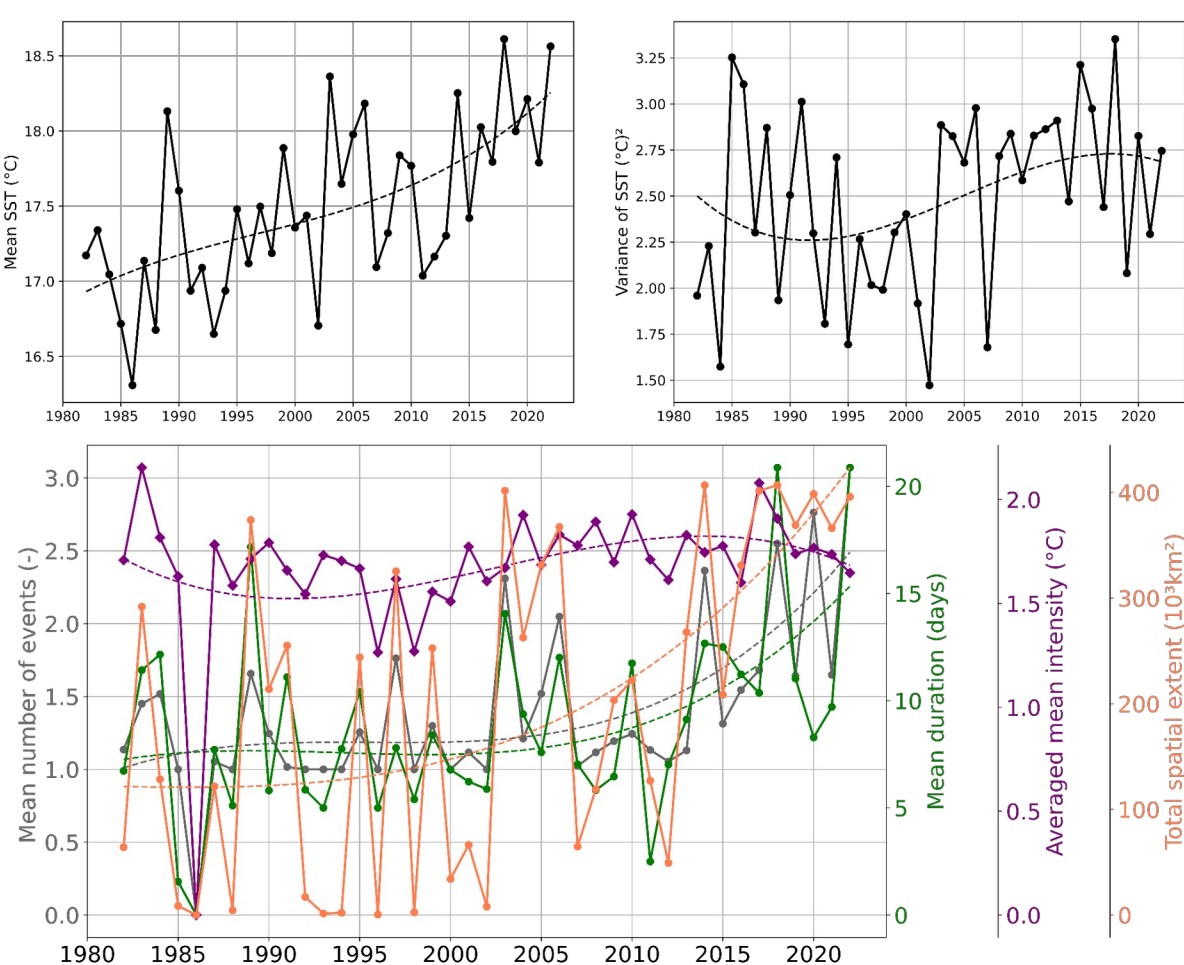

Figure 3: Time series of the mean (upper-left) and variance (upper-right) of SST (black curve)
of summers (JJAS) over the northeast Atlantic for the period 1982-2022. The SST variance is
calculated for each year over the respective domain and measures the spread of the spatial
distribution. (Bottom) Mean properties of summer (JJAS) MHWs in the northeast Atlantic.
The mean number of events (grey curve with circle marks) is the number of events within the
summer averaged over the domain (without considering cells with zero event). Mean duration
(green curve with circle marks) is the average duration of every event within the summer and
domain. Averaged mean intensity (purple curve with diamond marks) is the average of the
mean intensity of every event within the period and domain. Total spatial extent (orange curve
with circle marks) is the sum of each grid cell area where one or more events occur. If more
than one MHWs occurs on the same cell, only one grid cell area is taken into account. Dash
lines represent the regression of a third-order polynomial of the solid line with the same color.
The mean SST has been increasing over the 40 years with an approximately linear
trend, showing a mean warming of nearly 1.5 ºC for the whole domain since 1982 (Figure 3).
Regionally, it is observed that the increase in the mean SST is almost yearly constant for the
Bay of Biscay region, and quadratic for the English Channel and Bay of Brest, where a
plateau is observed around 1995-2010 (Figure S2).
The SST variance is calculated for each year over the respective domain and measures
the spread of the spatial distribution. Over the northeast Atlantic, during 1985-2002 and the 5
most recent years are characterized by a decline in the SST variance, while around 1992-2017
an increase in the SST variance is observed. This interannual trend is similar to the ones
observed for the events' intensity, with the exception of the English Channel, showing a direct
relationship between the SST variance and the mean intensity of the MHWs events. In the
English Channel, Bay of Brest and Bay of Biscay, the mean SST is warming and the variance
is  increasing. This estimation suggests that they both contribute to the changes in the
respective MHWs activity (Figure S2).
Contributing to this recent increase in the northeast Atlantic is primarily the sharp
trend of the events' spatial extent (~180 to 400 °C.days.$10^3$ km²), followed by the rise of the
number of events (1.2 to 2.5) and also their duration (7 to 15 days; Figure 3). One should note
that, for the same number of events, the events' spatial extent can differ depending on their
spatial repartition, as in the events' spatial extent only one grid cell area is taken into account
when more than one event occurs on the same grid. Furthermore, over the most recent years
the mean number of events, their mean duration and total spatial extent reached the maximum
recorded values. Since 2017, the total spatial extent over the northeast Atlantic has recorded
consecutive high values, exceeding 360 x$10^3$ km². The summers of 2018, 2020 and 2022
recorded on average more than 2.5 events for almost all subregions, with events lasting on
average more than 20 days in 2018 (Bay of Biscay) and 2022 (English Channel; Figure S2).
Among the three studied subregions, the English Channel is the region experiencing the
strongest increase in summer MHWs activity over the last four decades (see trend in Figure
S3). The longest mean duration is seen in the English Channel (35 days in summer 2022), the
highest mean number of events occurred in the Bay of Brest (2.7 in summer 2020) and the
highest mean intensity is present in the Bay of Biscay (2.2 ºC in 2017; Figure S2).
3.1.2. Coastal MHWs activity
The spatial heterogeneity of the MHWs occurrence and activity can influence the
impact of MHWs along the coastline. We now explore MHWs activity detected along the
coast from *in situ* observations compared with remotely sensed observations. Figure 4 shows
the activity detected for the whole northeast Atlantic domain and in the three subregions
where long-term *in situ* observations exist. To compare *in situ* and satellite data, for each
station, time series based on satellite data consider only years where in situ data exists (see
Table S1 for the starting date) and exceeds 80% of available data for the considered season.
Linked with the whole domain activity (Figure 4a), we observe an increase in the MHWs
activity in the three subregions (Figure 4b, c, d). Similar evolutions are observed when the
satellite product or coastal buoys are considered. In the Bay of Brest, we also observe a

similar increase but with larger activity in *in situ* observation as the intensity of extreme is underestimated by the satellite in this semi-enclosed bay. The use of *in situ* observation is limiting the length of the analyzed time series. However, we can observe larger activity in recent years from both datasets. For most cases, similar most active years are detected with *in situ* observations and satellite data.

Considering coastal stations over the observed periods, we see a more pronounced increase in MHWs activity from 2010. The English Channel and the Bay of Biscay *in situ* stations highlight the year 2022 as the most active year exceeding 140 °C.days. In the Bay of Brest, the impact of the 2022 MHWs is less pronounced, in agreement with satellite observation (Figure 2b) due to tidally driven vertical mixing.
When we compare MHWs activity estimated from *in situ* stations and satellite product, values are generally larger from *in situ* stations. Those differences are explained by the underestimation of extreme temperatures in coastal regions in remotely sensed products.

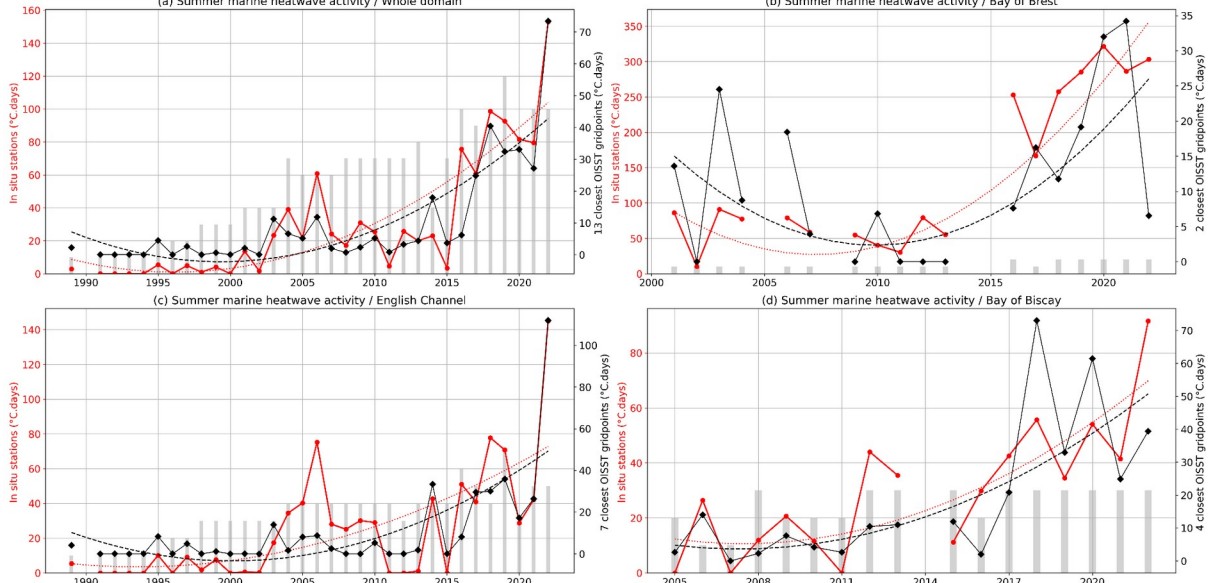

Figure 4: Time series of summer (JJAS) MHWs mean activity (a) in the whole domain (northeast Atlantic) and in three subregions: (b) Bay of Brest, (c) the English Channel, and (d) Bay of Biscay. The red curve with circle marks represents the activity based on *in situ* observations. The black curve with diamond marks represents the activity based on satellite dataset for the closest non-masked points with *in situ* stations when *in situ* data exists. Dash lines represent the regression of a third-order polynomial of the solid line with the same color. Grey bars are proportional to the number of considered *in situ* time series. MHWs activity from *in situ* time series with less than 80% of observation during the analyzed season is not computed.

3.2 Evolution of marine cold-spell activity

3.2.1 An integrated regional view

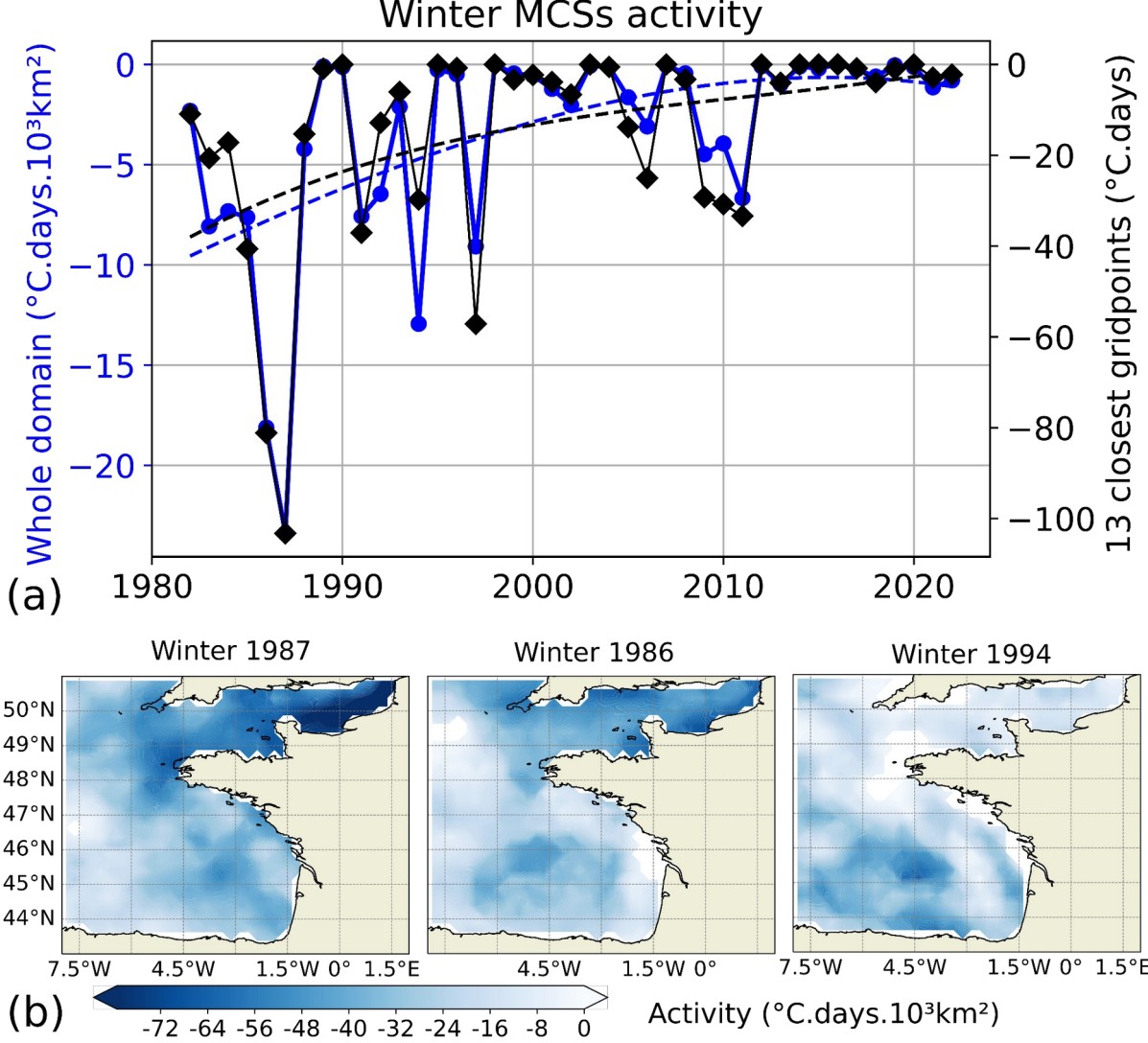

(a)

(b)

Figure 5: Same as Figure 2 but for MCSs in winter (DJFM).

Figure 5 depicts winter MCSs evolution for the whole domain over the last four decades (1982-2022). MCSs activities decrease linearly during the first half of the period, showing almost no occurrence after 2000 with the exception of 2006 and 2009 to 2011. A similar evolution is seen by considering the average of the 13 grid points closest to each *in situ* station.

The three most active MCSs occur in winter 1987 (-24 °C.days.$10^3$.km²), 1986 (-18 °C.days.10.km²) and 1994 (-13 °C.days.$10^3$.km²). In the two coldest winters, MCSs were dominant in the English Channel, especially off the northern French Coast in winter 1987. These two winters are characterized by long (~ 50 days) and intense (~ -2.5 ºC anomalous SST) and few events (~ 1 event; Figure S4). This region is subject to high turbulent mixing generated by the tidal current, which could favor cold conditions. By contrast to these two winters (1987 and 1986), winter 1994 featured strong MCSs activity in the center of the Bay of Biscay, due to numerous (~ 5 events) but moderate intensity (~ -1.3 ºC) and relatively short (20 days) events. The three winters 2009-2011 present very localized extreme cold conditions along the coastal Armorican Shelf, and additionally in the English Channel for 2011 (not shown). The detailed mean features (number of events, duration and mean intensity) of winter

MCSs over the period 1982-2022 in the northeast Atlantic, English Channel, Bay of Brest and
Bay of Biscay are documented in Table S3.

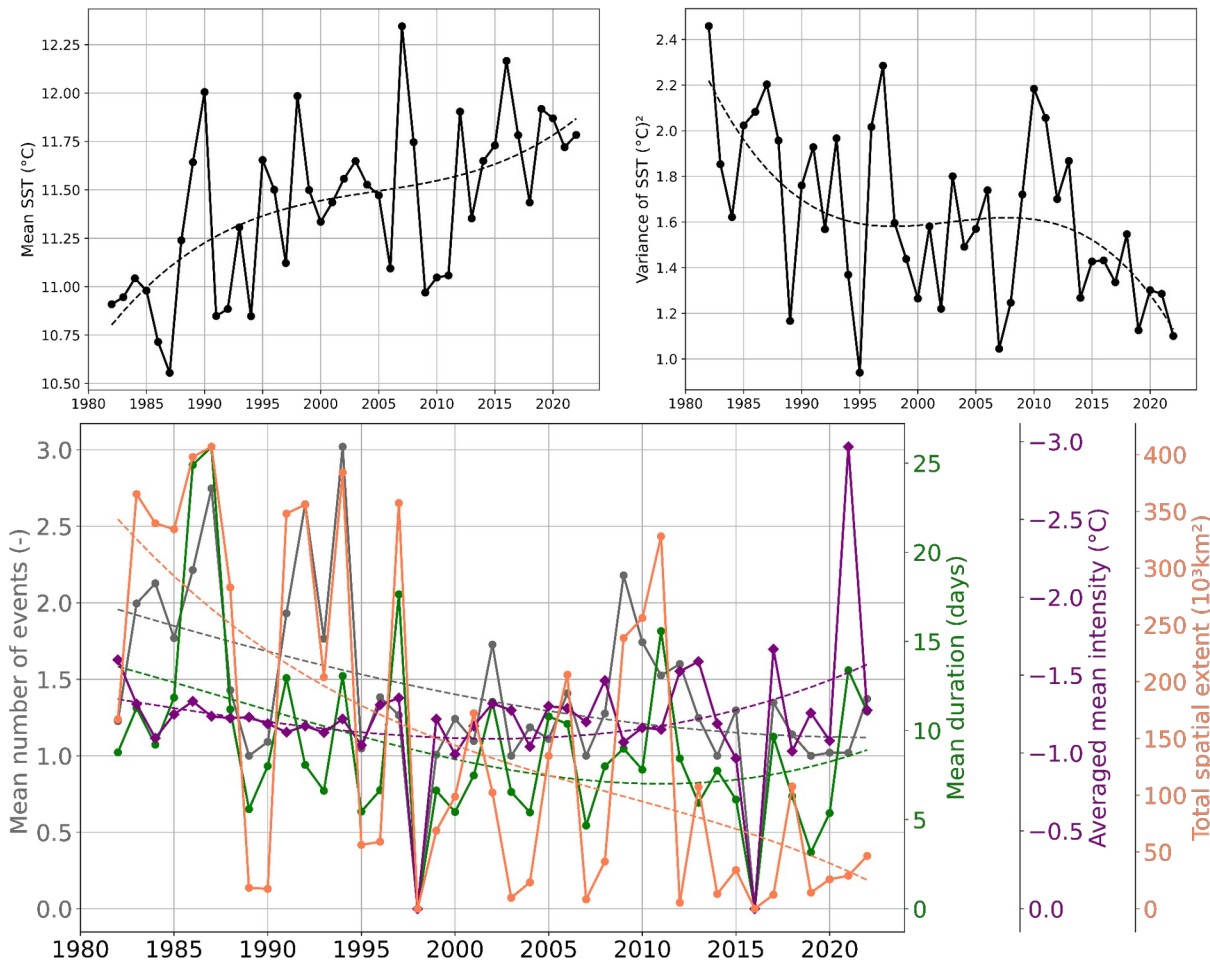

Figure 6: Same as Figure 3 but in winter (DJFM) and MCSs (blue curve)

The mean and variance evolution of SST, as well as the mean evolution of MCSs
properties (occurrence, duration, mean intensity and spatial extent) are presented over the
whole domain (Figure 6) and separately for the English Channel, the Bay of Brest and the Bay
of Biscay (Figure S5). Over the whole northeast Atlantic domain, the SST mean increases and
spatial dispersion (variance) decreases with both showing a plateau around 1995-2010,
following the English Channel and the Bay of Biscay evolution. On the contrary, a steady
increase in the mean SST and a nearly constant variance of SST is seen in the Bay of Brest.

The warmer winter seen over the whole domain and for the three subregions is
consistent with the decrease of the extremely cold conditions, depicted by the mean MCSs
activity. The decrease in the mean MCSs activity is controlled by the strong decrease in
spatial extent (350 to 50 x10³.km²), the moderate decrease in the number of events (2 to 1.2
events), and the small decrease in duration (13 to 9 days). The mean intensity does not show
any trend (~ -1.5 ºC).

The decrease of spatial dispersion (variance) of SST over the whole domain indicates
a more uniform evolution which is explained by a dominant warming trend stronger for colder
areas. Indeed, the relatively cold English Channel's temperature increased by 1.5 ºC (from 9
ºC to 10.5 ºC) and the relatively warmer Bay of Biscay increased by 0.8 ºC (from 11.8 ºC to
12.6 ºC) over the 1982-2022 period. When considering individually the three subregions,
localized enough to be under a similar trend, the variance also decreases (Figure S5). The
decrease of variance is more pronounced for the English Channel than for the Bay of Brest
and Bay of Biscay. Therefore, a first estimate shows that mean SST warming and the variance
changes both contribute to the changes in MCSs activity in the English Channel, Bay of Brest
and Bay of Biscay.

MCSs activity generally follows the SST evolution, albeit with small differences.
Indeed, winter 1991 and 1994 have a similar mean SST (10.8 ºC) but the MCSs activity is
three times higher in 1994 than in 1991, driven by a higher number of events (3 instead of 2
events with similar duration, mean intensity and spatial extent).

Even if changes in winter occur in the Bay of Brest and Bay of Biscay, more drastic
changes are seen in the English Channel over the period 1982-2022 (see trend in Figure S6).
In the English Channel, the trend of MCSs shows at the beginning of the period, a mean
occurrence of 2 events/winter, lasting 15 days with a mean intensity of -1.5 ºC over an area of
100 x10³ km², followed by a sharp decline ending to no detected MCSs in the last four years
(2019-2022). In the Bay of Brest over the same period, MCSs properties decrease from 1.5
events during 15 days at a mean intensity of -1.4 ºC over 11 x10³ km² to 0.5 events during 8
days at a mean intensity of -0.8 ºC over 0.5 x10³ km². Exceptional long events occurred in the
winter of 1987 with a mean duration of 55 days. In the Bay of Biscay, the MCSs decline in
occurrence (from 2 to 1 event), duration (from 11 to 9 days) and spatial extent (170 to 40 x10³
km²) while the mean intensity rises from -1.3 ºC to -1.5 ºC. The increase is explained by
winter 2021; without these events, the mean intensity would have been nearly constant around
-1.3 ºC. Indeed, winter 2021 shows a small activity but the highest mean intensity (-3 ºC over
the whole domain) which is explained by a localized event in the coastal area off
southwestern France with a maximum intensity of (-5.6 ºC). Apart from a very intense and
localized event in the coastal area off southwestern France in winter 2021 and a very long
event in the Bay of Brest in winter 1987, severe MCSs occurred predominantly in the English
Channel (winter 1987 and 1986).


3.2.2. Coastal MCSs activity

Figure 7 shows the time series of MCSs activity for *in situ* data and satellite data
considering the same missing data as each *in situ* station data. Along the coasts, MCSs
activity as determined by local buoys remains weaker than MHWs activity as defined using
satellite data. As for the MHWs, MCSs intensity is underestimated in satellite observations
but evolutions are similar. From *in situ* observations from coastal stations, two years can be
highlighted due to their intense MCSs: 2006 and 2010 (Figure 7). The year 2010 is the most
intense, in terms of MCSs. The mean activity is reaching -100 ºC.day in the Bay of Brest and
around -60 °C.day in the Bay of Biscay and the English Channel. In 2006, the activity was
also important compared with other years: around -80 °C.day in the Bay of Brest and around -
50 ºC.day in the English Channel. This extreme year 2006 was also unique with a peak in
MHWs activity during the summer (Figure 4). Before the year 2000, two other years reveal
intense MCSs activity in the coastal English Channel: 1997 and 1991 (from the most intense
to the less active winter).
We do not detect a significant trend in the interannual evolution of MCSs activity
along the coasts. For the Bay of Biscay and the Bay of Brest, it can be directly connected to
the lack of observation before 2000 when the largest MCS occurs. In the English Channel, the
lack of observation also explains the lack of a clear trend. Indeed, only one time series was
available before 1995 and this station (GREENwich) is not detecting an important MCSs
activity before 2000.

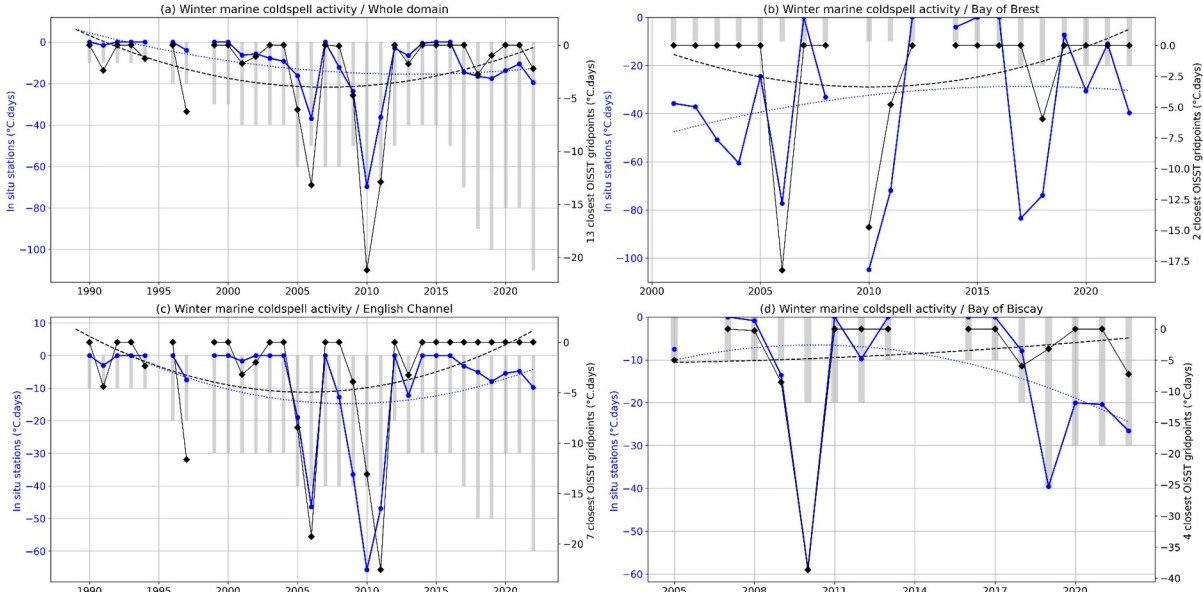

Figure 7: Same as Figure 4 but for MCSs in winter (DJFM).
3.3 Associated atmospheric patterns
Apart from the long-term trend of increasing SST, we also see high interannual
variability which is potentially connected with atmospheric forcing modes (Holbrook et al.
2019; Izquierdo et al., 2022a). Figure 8 presents the atmospheric circulation in the north
Atlantic associated with strong interannual MHWs in the Bay of Biscay and the English
Channel. For each summer of the 1982-2022 period, MHWs total activity anomaly in the
studied area box (northeast Atlantic) with respect to the third-order long-term trend (red
dotted curved in Figure 2a) was computed. This anomaly represents the detrended or
interannual MHWs activity. Eight summers were identified as having high interannual
activities (anomalous total activity exceeding a threshold of 4 °C.days.$10^6$.km², coloured
marker in Figure 8 left panel). The year 2018 (23 °C.days.$10^6$.km²), 2003 (17
°C.days.$10^6$.km²) and 2006 (12 °C.days.$10^6$.km²) are the three strongest summers. Six out of
these eight summers (all except 2018 and 2022) have an anomalous geopotential height at 500
hPa which is positive over northern Europe (box A in Figure 8) and negative west of the
Iberian Peninsula (box B in Figure 8). The composite of the anomalous geopotential height at
500 hPa for these six summers shows in the north Atlantic-Europe sector a positive summer
NAO-like pattern, with a high over the Nordic sea and two lows over the Iberian Peninsula
and Greenland. This overall result is not sensitive to small displacements of boxes (a few
latitude and longitude degrees; not shown).

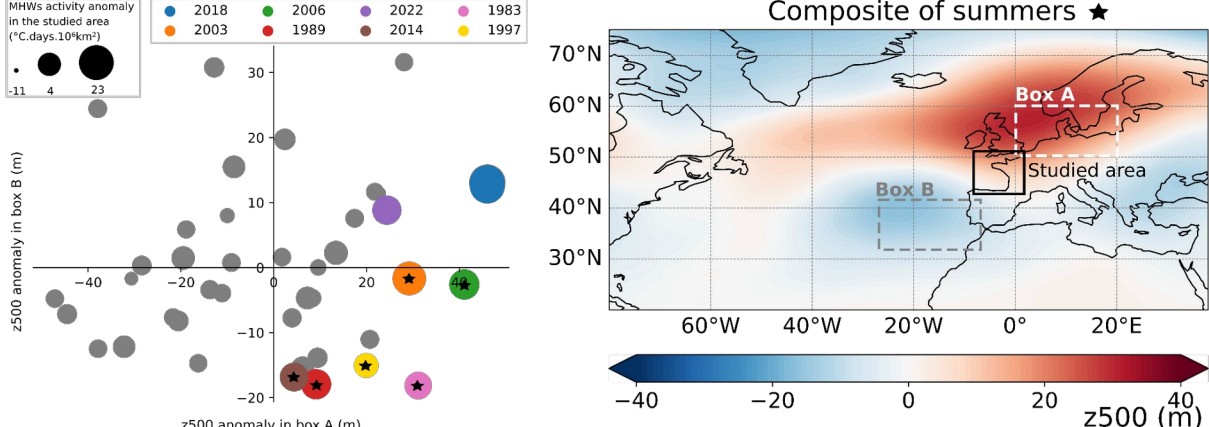

Figure 8: (Left panel) Scatter plot of anomalous summer (JJAS) geopotential height at 500 hPa (z500; in m) in box A versus the anomalous geopotential height at 500 hPa in box B with respect to the summer period 1982-2022. The size of the marker is proportional to the anomalous summer (JJAS) MHWs total activity, calculated as the sum of all grid point activity in the studied area (in °C.days.$10^6$.km²) with respect to the trend (red dotted curved in Figure 2a). Markers are in color when this value exceeds 4 °C.days.$10^6$.km² and the stars are indicated when markers in color are in the lower-right "cluster" of the graph. (Right panel) Composites of summers (JJAS) marked with stars in the left panel of the anomalous geopotential height at 500 hPa (m) with respect to the summer period 1982-2022. Box A is the domain 0°E to 20°E-50 °N to 60°N and box B is the domain 33°W to 13°W - 31°N to 41°N.

Summer (JJAS) 2018 has the strongest anomalous MHWs activity in the northeast Atlantic but, differently to the six next summers in the ranking of detrended MHWs activity, does not present a decrease in the geopotential height at 500 hPa west of the Iberian Peninsula (box B). A broad high-pressure system in the north Atlantic-European sector is seen (including box A), except in the eastern Mediterranean and to 60°N where a low occurs (Figure S7). This response in box B for summer 2018 is primarily due to late summer (August and September) atmospheric circulation (Figure S7). These months have a minor contribution to MHWs total activity for the whole summer (JJAS; Figure S8). When considering the month of June, with 2018 MHWs peaks (Figure S8), the north Atlantic shows a positive summer NAO regime, similar to the next six summers' highest MHWs activity. This analysis demonstrates that MHWs in the northeast Atlantic is closely associated with a high-pressure system over northern Europe, and a low off the Iberian Peninsula, resembling the positive phase of the summer NAO. By performing this analysis with SST instead of MHWs activity, we obtain similar results, albeit with a less extended high over northern Europe (Figure S9).

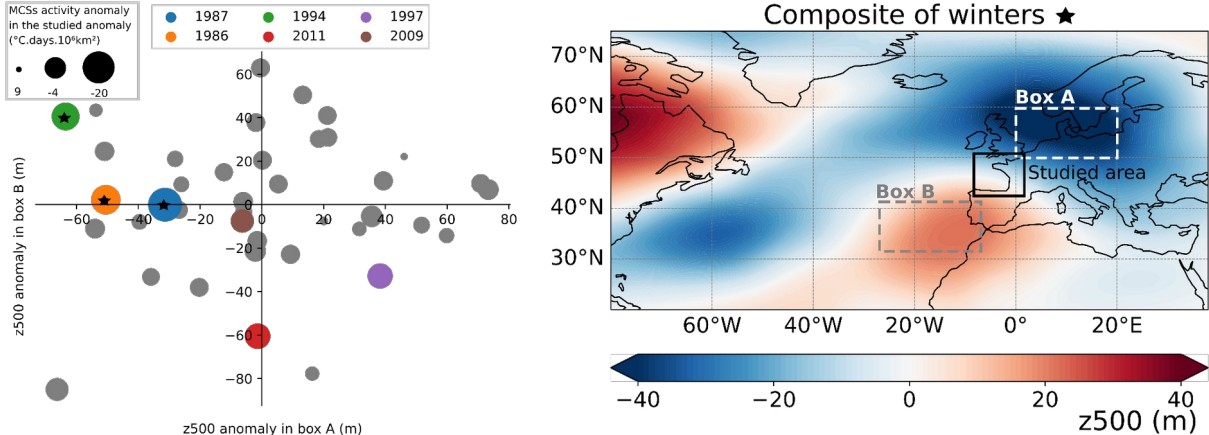

Figure 9: Same as Figure 8 but for MCSs in winter (DJFM). MCSs anomalies are calculated with respect to the third-order trend (blue dotted curved in Figure 5). Markers are in color when this value is below -4 °C.days.$10^6$.km² and stars are indicated when markers in color are in the upper-left section of the graph.

Regarding MCS, the three highest detrended MCSs activities are winter 1987 (-20 °C.days.$10^6$.km²), 1986 (-13 °C.days.$10^6$.km²) and 1994 (-10 °C.days.$10^6$.km²; Figure 9). These three most active winters are in the same "cluster", with anomalous 500 hPa geopotential height negative over northern Europe and positive west of the Iberian Peninsula. Composite of the anomalous geopotential height at 500 hPa for these three winters shows in the north Atlantic-Europe sector a broad and strong low in northern Europe, a weaker low-pressure system sitting in the northwest Atlantic, and two highs off the Iberian Peninsula and over the Hudson Bay. This analysis suggests that extreme MCSs in the northeast Atlantic might be closely associated with a low over northern Europe and a high off the Iberian Peninsula. By performing this analysis with SST instead of the MCSs activity (Figure S10), the result are sparse, showing only winter 1986 as strong anomalous cold SST linked to an anomalous geopotential height at 500 hPa over northern Europe and positive west of the Iberian Peninsula.

When comparing the anomalous geopotential height conditions for the most intense summer MHWs and winter MCS, we see that the geopotential height conditions are opposite, although the amplitude is stronger for winter, consistent with stronger climatology (Folland et al., 2009). However, while summer MHWs are associated with a positive summer NAO, winter MCSs are not associated with a negative winter NAO pattern.

To investigate potential drivers of these events, we have considered the different components of air-sea heat flux anomalies concomitant with MHWs and MCSs. For the eight most severe interannual summer MHWs (see marker in color Figure 8) and the six most severe interannual winter MCSs (see marker in color Figure 9), the anomalous (i) short-wave radiation flux, (ii) surface net long-wave radiation flux, (iii) surface sensible heat flux and (iv) latent heat flux are depicted, respectively Figure S11 and Figure S12. The interannual (or detrended) summer MHWs are predominantly driven by high short-wave radiation flux, except for years 1983 and 1997 that only shows important positive downward latent heat flux. The other air-sea fluxes have a smaller contribution. The interannual winter MCSs seem to be mostly driven by high sensible heat flux and low short-wave radiation flux. This suggests that, in this region, a decrease in cloud cover is a key parameter for the generation of summer MHWs while strong winds and an increase in cloud cover are important for the apparition of

winter MCSs. Further analysis needs to be done to attribute quantitatively the contribution of
each air-sea heat flux component.

## 4. Discussion

In the northeast Atlantic, an increase in the MHWs activity and a decrease in MCSs
activity were observed. Interannual changes confirm that general large scale trends (Oliver et
al., 2018; Schlegel et al., 2021) are also observed in regions where the coastal hydrodynamics
could limit the impact due to active vertical mixing processes (*e.g.* barotropic and internal
tides, wind-driven mixing in shallow waters).
The most active summer MHWs analyzed over the northeast Atlantic and in the period
1982-2022 occurred in the Bay of Biscay (2018) and the most active winter MCSs occurred in
the English Channel (1987). This is consistent with Schlegel et al. (2021) who found that the
maximum intensity of MHWs dominates MCSs in the Bay of Biscay, and vice versa in the
English Channel. Along the coasts, the maximum of MHWs activity is detected in 2022 in the
English Channel which might be related to the summer European heatwaves recorded
(ECMWF, 2022; Savu, 2022; Guinaldo et al., 2023).
In the Bay of Biscay, we see a linear warming rate in summer since the beginning of
the studied period. This is in accordance with DeCastro et al. (2009) which shows a steady
linear warming rate since the 1970s, based on data from 1854-2006. Mean SST together with
SST variance increase may explain the increase of MHW. This increase of MHWs is
consistent with Izquierdo et al. (2022a) who determined more precisely an equal contribution
of each of these two factors for the South coast of the Bay of Biscay. This is specific to this
region (as well as for the Bay of Brest and the English Channel), as for most of the other
regions of the world, the mean warming and not the SST variability changes contribute to the
increase in MHWs features (Alexander et al., 2018; Oliver et al., 2020). In addition, we found
a positive trend for the MHWs activity parameter using both satellite data and the 4 buoys in
the Bay of Biscay, and for the duration and occurrence using satellite data. The trends are
quasi-similar considering only the two buoys on the South coast of the Bay of Biscay (GIJO
and BILB) and the two on the west coast of the Bay of Biscay (ARCA and MOLI; not shown)
and are marked by the high activity present in the more recent summers. This evolution in the
occurrence and duration of MHWs were not seen in Izquierdo et al. (2022b) using two buoys
in the South coastal Bay of Biscay over the period 1998-2018, which could be explained by
local process or studied season (March to August).
The results from *in situ* and satellite datasets for each of the studied regions are quite
in agreement, albeit the satellite underestimates the amplitude of activity for both MHWs and
MCS. Conversely, Izquierdo et al. (2022a) found an overestimation of the MHWs using
satellites compared to *in situ* in the coastal upwelling region South of the Bay of Biscay,
which might be related to local processes. The satellite's coarse resolution mostly (i) smoothes
small-scale and short events and (ii) interpolates with offshore regions, having greater thermal
inertia (Marin et al., 2021) which can lead to the overestimation of the duration of events and
the underestimation of the intensity. However, we show that coastal *in situ* stations distributed
along the northeast Atlantic coasts allow the detection of large-scale evolutions of MHWs and
MCSs activity. Analyzed locally, they can also inform on evolutions related to local
hydrodynamics.
Internal variability of winter MCSs is related to low pressure over northern Europe
and a high-pressure West of the Iberian Peninsula for three (1987, 1986 and 1994) out of the
six most intense events. Among other strong interannual MCSs, winter 2011 does not present
this pattern but could have been generated by a cold air outbreak brought by a ridge over
Greenland (Norris et al., 2013).   A relation at an interannual timescale could exist between
MCSs (Figure 7, middle panel) and extreme low-salinity events (Poppeschi et al., 2021) in
winter in the Bay of Brest, as, using the same *in situ* buoys (COAST-HF-Iroise from 2000-
2018), two out of the four most severe low-salinity events are concomitant with MCSs (winter
2001 and 2007). These extreme events could be both influenced by intense mid-latitude
depressions, but river discharges are also an important driver in this region. Unlike MHWs
(Figure 2), extreme cold conditions occurred several winters in a row: three in 2009-2011 and
two in 1986-1987. This might be explained by the re-emergence of cold water originating
from the previous winter, as for the 2013-2016 north Atlantic cold Blob (Duchez et al., 2016a;
Josey et al., 2018; Schlegel et al., 2021).
Summer 2018 presents the most active MHWs in the northeast Atlantic for the period
1982-2022, consistent with the reported warmer SST the same summer (+1 to +3 °C above
the long-term climatology) in the proximity of the United Kingdom (McCarty et al., 2019).
Over land, this summer was also recorded as the hottest in the United Kingdom since 1884
(McCarty et al., 2019) and one of the hottest over northwestern Europe (Met Office, 2018;
Météo-France, 2018). On top of the underlying warming climate forcing (Vogel et al., 2019;
Yiou et al., 2020), this extreme continental warm conditions in 2018 have been previously
reported as a consequence of the positive summer NAO anomalies combined with elevated
SST (McCarty et al., 2019) or combined with stationary Rossby waves in synoptic anomalies
(Drouard et al., 2019; Kornhuber et al., 2019). More generally, the positive phase of the
summer NAO is associated with warm anomalies from the west of the United Kingdom to the
Baltic (Folland et al., 2009). Our findings for MHWs corroborate the continental counterpart
as extremely warm conditions in the Bay of Biscay and the English Channel are likely
associated with positive summer NAO, consistent with the result of Holbrook et al. (2019).
Depending on the region and the event, MHWs can be associated with anomalous air–
sea heat fluxes which can include high short-wave fluxes, due to less cloud cover and greater
insolation, high sensible heat fluxes when the surface air is warm and/or low latent heat loss
from the ocean, due to weak winds (Oliver et al., 2021). In the English Channel and the Bay
of Biscay, Guinaldo et al. (2023) linked the summer of 2022 sea-surface temperature to
abnormally high short-wave radiation in the Bay of Biscay and English Channel. In this study,
a similar conclusion is found by considering the eight most severe interannual MHWs in the
northeast Atlantic (which includes the English Channel and the Bay of Biscay, and summer of
2022). Abnormally high short-wave radiation is likely associated with reduced cloudiness and
Folland et al. (2009) have found that during the positive index phase of the summer NAO,
northwest Europe experiences significantly reduced cloudiness. This is consistent with our
suggestion that the positive phase of the summer NAO favours the generations of summer
MHWs in the northeast Atlantic through reduced cloudiness. MCSs in the English Channel
are associated with high sensible heat fluxes, consistent with reported MCSs often driven by
strong winds in shallow waters, enabling a rapid cooling of the surface water (Crisp, 1964;
Schlegel et al., 2021). We also found a possible role of weaker short-wave radiation, which
might be related to increased cloud coverage.
In the future and under increasing greenhouse gas concentrations, climate models
predict that the ocean surface in the Bay of Biscay and the English Channel will continue to
warm (Fox-Kemper et al., 2021) and a trend toward a positive summer NAO pattern (Faranda
et al., 2019). Both these effects imply the long-term likelihood of increased MHWs in the
northeast Atlantic, but to what extent are the long-term and the interannual variability
contributions remain to be shown. Also, the role of large-scale ocean circulation features,
such as the Shelf Edge Current (Alheit et al., 2019) or Iberian Poleward Current (Charria et
al., 2013), upper ocean preconditioning (Josey et al., 2018), and the importance of remote
large-scale climate modes of variability, such as the Indian Ocean Dipole (Holbrook et al.,
2019) in amplifying or suppressing MHWs occurrences in the Bay of Biscay and English
Channel would need specific investigation. Along the coasts, the role of main river inflow at
the land-sea continuum can also lead to specific answers on the coastal ocean to future climate
evolutions.

## 5. Conclusions

The activity index, a combination of the properties of marine extreme events, shows a
positive trend for summer MHWs in the northeast Atlantic (since 2000 and more pronounced
since 2010) and in the three subregions, the English Channel, the Bay of Brest and the Bay of
Biscay for both *in situ* and satellite data. This is explained by both a mean and variance SST
increase. Conversely, a decrease in MCSs activity was detected, with almost no events after
2000, more clearly shown with the satellite data due to the longer time series (40 years)
compared with the *in situ* (20 to 30 years). These changes are fast for the three subregions,
with the English Channel being the subregion with the more drastic growth.
In the northeast Atlantic, MHWs are more frequent, longer, and extend over larger areas,
while the opposite is seen for MCSs. For both MHWs and MCSs, the mean intensity shows
only weak changes over the last four decades.
Moreover, we found that the satellite dataset used is in good accordance with *in situ* data in
the northeast Atlantic, except for the fact that satellites underestimate the amplitude of both
hot summer and cold winter marine extreme events in the coastal areas. The implemented *in*
*situ* stations appear as a well-designed observing system to detect the long-term evolution of
MHWs and MCSs activity and to document local features related to coastal hydrodynamics.

MHWs activity is particularly high in 2018 and 2022 through two different situations.
The year 2018 is characterized by a large extent of MHWs in the Bay of Biscay with long
events in the South of the Bay and intense events in the Armorican Shelf. The summer of
2022 features long MHWs mainly in the English Channel. MCSs activity is the highest in
1986 and 1987 due to long and intense events in the English Channel.
Our findings show that summers with strong MHWs activity due to internal variability
(after removing the trend) in northeast Atlantic have often been associated with a ridge over
the northern Europe sea and a trough west of the Iberian Peninsula; the opposite situation is
seen for MCSs. In the case of MHW, the wide atmospheric pattern resembles the positive
phase of the summer NAO. This preliminary analysis of air-sea heat fluxes suggests that in
the northeast Atlantic interannual (or detrended) summer MHWs are predominantly driven by
high short-wave radiation flux and interannual winter MCSs by high sensible heat flux and
low short-wave radiation. This suggests that, in this region, decreased cloud cover is a key
parameter for the generation of summer MHWs while strong winds and increased cloud cover
is important for the apparition of winter MCSs. We caution that the proposed connection does
not necessarily indicate causal links but these relations can provide indications of drivers.

Despite contrasting hydrodynamical regimes (meso- and macro-tidal) and circulation
(shallow water under freshwater influence, shelf circulation, active sub-mesoscale), the
northeast Atlantic region displays similar changes in MHWs and MCSs activity between
coastal and open ocean regions. Those changes need to be anticipated to mitigate the impacts
on coastal ecosystems.

# Acknowledgements

This work was partially supported by national funds through FCT (Fundação para a
Ciência e a Tecnologia, Portugal) through project ROADMAP (JPIOCEANS/0001/2019). It
is also funded by the regional project (Contrat Plan Etat-Region) ObsOcean/ROEC-ILICO
and the regional COXTCLIM project funded by the Loire-Brittany Water Agency, the
Brittany region, and Ifremer. We thank Oregan Segalen for fruitful discussions. We thank
Tim Smyth for providing data from Western Channel Observatory. We acknowledge the
COAST-HF (http://www.coast-hf.fr) national observing network component of the National
Research Infrastructure ILICO.   Additionally, we express our gratitude to the two reviewers
and editor for their valuable feedback, which has greatly contributed to improving the quality
of our work.

# Declaration of competing interest

The authors declare that they have no known competing financial interests or personal
relationships that could have appeared to influence the work reported in this paper.
# Authors contributions

All authors contributed to the conception and design of the study. AS performed the
calculation and designed the figures involving the satellite dataset, GC and CP did so for the
*in situ* dataset. All authors contributed to the discussion, writing and review of the manuscript.

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
