# Peer review of "Coastal and regional marine heatwaves and cold-spells in the Northeast Atlantic"

_EGUsphere, 2023_

## Author Response (AR1)

**Coastal and regional marine heatwaves and cold-spells in the Northeast Atlantic**

**REVIEW REPORT - OCEAN SCIENCE**

**Review #1**

This study uses satellite data and coast mooring observations to detect summer marine heatwaves and winter cold spells in 3 coastal regions in the northeast Atlantic. Summer marine heatwaves are more frequent, longer, and extended over a larger area over the past decades and the marine cold spells have opposite trends. It is speculated that the high/low-pressure systems in the region are key drivers of extreme events, especially the marine heatwaves in 2003, 2018, and 2022 and their spatial distributions. In general, this is an interesting study to provide background information for a better understanding of the climate drivers and future climate projections of these extreme events. Here are some suggested revisions before the manuscript can be considered for publication.

The main comment is on the atmospheric patterns associated with the marine heatwaves and cold spells. It is nice to show the geopotential height anomalies associated with the events, however, it would be more informative to be quantitative about the drivers of the events. It is preferred to have a mixed-layer heat budget analysis, or at least show the different components of air-sea heat flux anomalies, which would provide some indication of the drivers of the events.

*Thank you very much for this interesting point. We have now extended our analysis by considering the different components of air-sea heat flux anomalies for marine heatwaves and marine cold-spells, which provides indication on the drivers of the events. We have added two related figures in the supplementary file showing for the eight most severe interannual summer MHWs (Figure S11) and the six most severe interannual winter MCSs (Figure S12), the surface net short-wave radiation flux, surface net long-wave radiation flux, surface sensible heat flux and latent heat flux.*

*We now added in the abstract: "A preliminary analysis of air-sea heat flux suggests that, in this region, low cloud coverage is a key parameter for the generation of summer MHWs while strong winds and high cloud coverage is important for the apparition of winter MCSs."*

*We have added in the methodology part: "Monthly geopotential height at 500 hPa (Z500), surface net short-wave radiation flux, surface net long-wave radiation flux, surface sensible heat flux and latent heat flux data were obtained from the European Center for Medium-Range Weather Forecasts (ECMWF) reanalysis data ERA5 at a spatial resolution of 0.25°× 0.25° (Hersbach et al., 2019)".*

*We now say in the result part: "To provide indications on the drivers of these events, we have considered the different components of air-sea heat flux anomalies concomitant with MHWs and MCSs. For the eight most severe interannual summer MHWs (see marker in color Figure 8) and the six most severe interannual winter MCSs (see marker in color Figure 9), the anomalous (i) short-wave radiation flux, (ii) surface net long-wave radiation flux, (iii) surface sensible heat flux and (iv) latent heat flux are depicted, respectively Figure S11 and Figure S12. The interannual (or detrended) summer MHWs are predominantly driven by*

*high short-wave radiation flux, except for years 1983 and 1997 that only shows important positive downward latent heat flux. The other air-sea flux have a smaller contribution. The interannual winter MCSs seem to be mostly driven by high sensible heat flux and low short-wave radiation flux. This suggests that, in this region, low cloud cover is a key parameter for the generation of summer MHWs while strong winds and high cloud cover are important for the apparition of winter MCSs. Further analysis needs to be done to attribute quantitatively the contribution of each air-sea heat flux component."*

*We now say in the discussion part: "Depending on the region and the event, MHWs can be associated with anomalous air–sea heat fluxes which can include high short-wave, due to less cloud cover and greater insolation, high sensible heat fluxes when the surface air is warm and/or low latent heat loss from the ocean, due to weak winds (Oliver et al., 2021). In the English Channel and the Bay of Biscay, Guinaldo et al. (2023) linked the summer of 2022 sea-surface temperature to abnormally high short-wave radiation in the Bay of Biscay and English Channel. In this study, a similar conclusion is found by considering the eight most severe interannual MHWs in the Northeast Atlantic (which includes the English Channel and the Bay of Biscay, and summer of 2022). Abnormally high short-wave radiation is likely associated with reduced cloudiness and Folland et al. (2009) have found that during the positive index phase of the summer NAO, northwest Europe experiences significantly reduced cloudiness. This is consistent with our suggestion that the positive phase of the summer NAO favours the generations of summer MHWs in the Northeast Atlantic through reduced cloudiness. MCSs in the English Channel are associated with high sensible heat flux, consistent with reported MCSs often driven by strong winds in shallow waters, enabling a rapid chilling of the surface water (Crisp, 1964; Schlegel et al., 2021). We also found a possible role of weaker short-wave radiation, which might be related to increased cloud coverage."*

*We now added in the conclusion: "This preliminary analysis of air-sea heat flux suggests that in the Northeast Atlantic interannual (or detrended) summer MHWs are predominantly driven by high short-wave radiation flux and interannual winter MCSs by high sensible heat flux and low short-wave radiation. This suggests that, in this region, low cloud cover is a key parameter for the generation of summer MHWs while strong winds and high cloud cover is important for the apparition of winter MCSs."*

*Hersbach, H., Bell, B., Berrisford, P., Hirahara, S., Horányi, A., Muñoz-Sabater, J., ... & Thépaut, J. N. (2020). The ERA5 global reanalysis. Quarterly Journal of the Royal Meteorological Society, 146(730), 1999-2049. (Accessed on 25-05-2023)*

From your mooring observations, have you observed the vertical extent of the heatwave and cold spell signatures?

*In situ observations allowing to observe the vertical extent of MHW or MCS were not available. Further studies are planned to explore the propagation of heatwaves in the water column from numerical simulations. Those simulations are now under development and validation, and we expect to be able to use them in the near future to investigate the vertical extent of MHW/MCS.*

It would also be good to know how the upper ocean in the region would precondition the marine heatwaves in the summertime, in addition to the atmospheric forcing.

*We agree with the suggestion but we opted here by only focusing on atmospheric forcing. Indeed, this first study focuses on in-phase and surface properties for in situ and satellite data, so we think the investigation of preconditioning is out of the scope. However, this was added as future work.*

*We now say: "Also, the role of large-scale ocean circulation features, such as the Shelf Edge Current (Alheit et al., 2019) or Iberian Poleward Current (Charria et al., 2013), upper ocean preconditioning (Josey et al., 2018), and the importance of remote large-scale climate modes of variability, such as the Indian Ocean Dipole (Holbrook et al., 2019) in amplifying or suppressing MHW occurrences in the Bay of Biscay and English Channel would need specific investigation."*

Here are some specific comments.

The writing needs to be improved. Here are just some examples:

Line 72: "significant difference" – not informative

*We now say "strong anomaly".*

Line 73: "more or less extensive" – not informative

*We now say: "which can occur at regional spatial scale"*

Line 231-236: the equation is not clearly explained.

*We acknowledge the reviewer's concern and we have added: "The activity is calculated for each grid point. It sums the product of the mean intensity, duration within the selected time range, and area of each detected event occurring within the selected time range."*

Line 316: "yearly constant" – wording

*We have removed "yearly". We now say: "Regionally, it is observed that the increase in the mean SST is almost constant for the Bay of Biscay region…"*

Line 319: "the first 10 years" – not actually. The smoothing distorted the decadal variations. The decline is during 1985-2002 from the yearly data.

*Thanks. We now say "during 1985-2002" instead of the first 10 years.*

Line 321: "spatial dispersion" is not defined.

*Thanks. We have replaced it with "variance".*

Line 323: how is the SST variance calculated?

*This information was missing in the document. A sentence was added: "The SST variance is calculated for each year over the respective domain and measures the spread of the spatial distribution."*

**Review #2**
This is a very interesting and detailed study on changes in seawater temperatures and extreme events (heatwaves and cold spells) in the NE Atlantic. The results are clearly presented and the methods are robust and based on previous research. The

characterization of extreme temperature events and their relationship with atmospheric patterns is also of interest for understanding larger-scale connections. I have some comments on a few aspects that are less clear or detailed in the preprint.

1.- (line 234): although the authors refer to Simmons et al., 2022, nor there nor in this preprint there is a detailed explanation about the determination of area for these marine extreme events.

*We agree with the reviewer that this was not clear and the following sentences was added: "and areaEE (in km2) is the area affected by the discrete event within a predefined domain". We have chosen three domains related to seas with different hydrodynamics (The English Channel, the Bay of Brest and the Bay of Biscay).*

*"Those three subregions can be associated with three contrasted hydrodynamical regimes: macrotidal (English Channel), semi-enclosed bay (Bay of Brest), mesotidal (Bay of Biscay; Charria et al., 2013)."*

2.- Figures 2(a) and 5 (a) depict the time series of the extreme events as yearly mean. There are two issues for this representation; the first is the number of data used for the calculation of that mean values, which is not given in the M&M section; and related to that, the graphs in figures 3 (upper right) and 6 (upper right) show the variance. Again, no indication of the number of data, and there are very significant variations, indicating that the dispersion of data can be very large in some cases, with variances even exceeding in two orders of magnitude the mean value.

*We agree with the reviewer that this needs to be explained better. So the number of data used depends on the number of events detected and is shown in the Figure 2(c) and 5(c). We removed the variance of the extreme events previously represented on Figure 3 (upper right) and Figure 6 (upper right). We also made it clearer how we calculated the SST variance.*

*We now say in the caption of Figure 2 and 5: "The SST variance is calculated for each year over the respective domain and measures the spread of the spatial distribution."*

3.- Also in the legends to figures 3 and 6, please change 'middle' by upper right, otherwise is a bit confusing.

*It has been corrected.*

4.- Figures 2b and 5b show the spatial distribution, along the whole domain, of the extreme events, but note that the 'whole domain', as depicted in figure 1 (left), does not include the central portion of the Bay of Biscay. If results have been interpolated, this should be clearly explained. The data points shown in figure 1(right) do not include this area.

*Thank you. We have now changed Figure 1 (left) to clarify what we mean by "whole domain".*

5.- Figures 4 and 7 (both lower right), please eliminate decimal scale for the x axis (year).

*Figures 4 and 7 have been updated following referee comments.*